# Multifidelity Model Calibration in Structural Dynamics Using Stochastic Variational Inference on Manifolds

**DOI:** 10.3390/e24091291

**Published:** 2022-09-13

**Authors:** Panagiotis Tsilifis, Piyush Pandita, Sayan Ghosh, Liping Wang

**Affiliations:** Probabilistic Design Group, General Electric Research, Niskayuna, NY 12309, USA

**Keywords:** Gaussian processes, stochastic variational inference, multifidelity modeling, manifold gradient ascent, structural dynamics, vibration torsion

## Abstract

Bayesian techniques for engineering problems, which rely on Gaussian process (GP) regression, are known for their ability to quantify epistemic and aleatory uncertainties and for being data efficient. The mathematical elegance of applying these methods usually comes at a high computational cost when compared to deterministic and empirical Bayesian methods. Furthermore, using these methods becomes practically infeasible in scenarios characterized by a large number of inputs and thousands of training data. The focus of this work is on enhancing Gaussian process based metamodeling and model calibration tasks, when the size of the training datasets is significantly large. To achieve this goal, we employ a stochastic variational inference algorithm that enables rapid statistical learning of the calibration parameters and hyperparameter tuning, while retaining the rigor of Bayesian inference. The numerical performance of the algorithm is demonstrated on multiple metamodeling and model calibration problems with thousands of training data.

## 1. Introduction

Modern engineering tasks are often characterized by the need to perform large scale expensive laboratory experiments or amortize hours of computation, performing simulations that are based on sophisticated mathematical formulations. While these “high-fidelity” sources of information provide detailed insight into the complex physical process, one usually faces a heavy computational runtime or a massive financial investment. In addition to this, obtaining data by running experiments or simulations needs more advanced insight, that might not always be extricated from the data by applying state-of-the-art methods used to build data-driven metamodels [1]. Finally, with the advent of Industry 4.0 [2], developing digital twins, which are commonly probabilistic surrogate models representing the underlying physical process, is becoming a routine practice across the industry. In a realistic scenario, the paucity of data and noise in the recorded measurements are challenges that also need to be taken into account.

Surrogate modeling methods that have shown promise in dealing with problems of the aforementioned kind, typically include Gaussian process (GP) regression [3,4,5], probabilistic deep neural networks [6,7,8] or polynomial chaos expansions [9,10,11]. The application of these methods has been extended to problems from different domains, such as manufacturing [12,13], flow through porous media [10,14], and combustion mechanics [15]. Classic formulations of these methods provide a meaningful representation of a model from uncertainty and noise, and they demonstrate strong predictive performance on unseen data. However, these approaches are susceptible to challenges such as limited training data, multiple sources of information that model the same process, and the lack of identifiability of model parameters [16]. By referring to the model at-hand that meets the accuracy required by the current application as a “high-fidelity” model, one standard approach in the literature is to employ similar “low-fidelity” models whose characteristic is that they provide a lower accuracy on the model output, though they are cheaper to evaluate. A detailed review on multifidelity approaches, that is, approaches that employ more than one models to approximate the same physical process with different levels of accuracy, can be found in [17]. In brief, the most common ways in which the low accuracy–low computational cost is achieved are by simplified physics models (coarse-grid PDE solver) [18], reduced-order models [19], data-fit interpolation models [20,21] and machine learning and mathematical surrogates [22,23].

In this work, our focus is on applying GP regression to problems that have thousands of data [24]. Secondly, we focus on the use of GP regression in both the single-fidelity (where only a high-fidelity model is considered) and the multifidelity (both a low- and a high-fidelity model are employed) modeling scenarios. In the second scenario, we focus on the case where data from two sources of varying fidelity are available, and the task involves calibrating the so-called tuners of the low-fidelity source. In all these tasks, we resort to a fully Bayesian formulation of the GP regression, differentiating ourselves from the works of [25,26,27], the details of which are discussed in Ghosh et al. [28]. This is a critical aspect of this work, as retaining a fully Bayesian treatment for the metamodeling and model calibration tasks with GPs is a major challenge from a computational and numerical perspective. In some of the authors’ previous work (see Pandita et al. [29]), it was demonstrated how savings in computational time could be achieved using adaptive sequential Monte Carlo methods fused with a fully Bayesian treatment, applied to tasks of the above kind. However, the utilization of hundreds of computational processing units or cores is not always practically possible, necessitating the need for alternative approaches. Other adaptive algorithms that accelerate Markov chain Monte Carlo methods for Bayesian inference [30,31,32] and optimal-transport-based approaches that circumvent the need for MCMC methods [33] have shown promise in recent years.

Most of the above-mentioned works rely on computational power and heavy use of large-scale computing in order to overcome the challenges of training the models. Our main contribution in this work is to achieve computational efficiency by leveraging a variational formulation of Bayesian inference, commonly known as black-box variational inference (BBVI) [34], and by improving the performance of the optimization scheme involved using efficient subsampling, rather than resorting to online access to exorbitant computational resources.

Variational methods [35,36] to Bayesian inference have shown promise in various tasks that resort to a Bayesian formalism in order to train surrogate models [37,38], calibrate physical models [39], and more recently across a swathe of deep learning tasks [40,41,42]. The key ingredient in variational inference (VI) that enables efficient posterior density exploration conditioned on large quantities of data is to perform the required likelihood function evaluations using random batch-sampling. Introducing this additional level of stochasticity in the algorithm, resulting in what is known as stochastic variational inference (SVI) [43], allows for fast likelihood evaluations during the optimization procedure and scales the algorithm, while a full exploration of the available training dataset is still guaranteed. SVI has been previously successfully applied for training deep GP models [44] and sparse GPs in big data scenarios [45]. In this work, we apply SVI to train hybrid Gaussian process models that make use of training data stemming from multiple levels of fidelity, while at the same time they can incorporate calibration parameters. Specifically, we adopt the well-known Kennedy–O’Hagan formulation [46] that relies on an autoregressive GP scheme, and we develop a training algorithm that scales BBVI for big data problems using batch-sampling. We identify the optimal Gaussian approximations to the true posterior densities of the model’s hyperparameters by solving the variational problem with respect to full covariance matrices, thus capturing all correlations between the parameters. To achieve this, we make use of a manifold gradient ascent algorithm that performs the optimization directly on the manifold of symmetric positive semidefinite matrices, as opposed to solving complex constrained optimization problems.

The outline of the paper is as follows: We present the mathematical details of the autoregressive multifidelity calibration model in Section 2. In Section 3 and Section 4, we expand on the details of the black-box variational inference and its use in scaling up for big data problems, and we introduce the manifold gradient ascent optimization scheme, to be used for carrying out the optimization task. To illustrate the direct applicability of the proposed approach on calibrating models using data from sources of varying fidelity, we use a set of synthetic functions in Section 5.1. We demonstrate the impact of the extended variational formulation on a benchmark machine learning dataset with thousands of training data, in Section 5.2. In Section 5.3, we highlight the impact of the proposed formulation on a challenging multifidelity problem, in the high-sample regime with over ten thousand training data, where the parameters of interest include the uncertain tuners of the low-fidelity simulation model. We summarize our conclusions and directions for future work in Section 6.

## 2. Multifidelity Gaussian Process Modeling and Calibration

### 2.1. Autoregressive Gaussian Processes

We consider the Kennedy and O’Hagan’s formulation [47], where two simulators are available, namely $y_{h}\left(x\right),y_{l}\left(x,\theta \right)$, where $y_{h}$ represents some high-fidelity computer code and $y_{l}\left(x,\theta \right)$ represents a low-fidelity simulation code. The design variable x is assumed to take values within a space of feasible designs $X\subset R^{D}$, while θ is a set of calibration parameters that characterize the low-fidelity simulator.

The relationship between the two codes is assumed to be
(1)$$y_{h}\left(x,\theta \right)=\rho y_{l}\left(x,\theta \right)+\delta \left(x\right)+\epsilon \left(x\right),$$
where δ(x) is a discrepancy term that is statistically independent of $y_{l}\left(x,\theta \right)$ and ϵ(x) accounts for measurement noise and is independent of both $y_{l}\left(x,\theta \right)$ and δ(x). The coefficient ρ satisfies
(2)$$\rho =\frac{\mathrm{cov}[y_{h}\left(x,\theta \right),y_{l}\left(x,\theta \right)]}{\mathrm{var}[y_{l}\left(x,\theta \right)]}$$
and therefore accounts for the correlation between the models. Although in general, ρ can be considered a function of x [48,49], we assume for simplicity that it is constant throughout this work. Further, we take $y_{l}\left(x,\theta \right)$, δ(x) to be Gaussian processes with zero mean and variances $\sigma_{l}^{2}r_{l}\left(x,x^{\prime }\right)$ and $\sigma_{\delta }^{2}r_{\delta }\left(x,x^{\prime }\right)$, respectively, where $r_{l}$ and $r_{\delta }$ are correlation kernels, here to be taken as squared exponential functions
(3)$$r_{t}\left(x,x^{\prime }\right)=\exp\left[-\sum_{i=1}^{D}\frac{{\left(x_{i}-x_{i}^{\prime }\right)}^{2}}{\ell_{i,t}^{2}}\right],\;\;t=l,\delta ,$$
with $\ell_{i,t}$ being the correlation length or length scale along dimension *i*, for the two kernels (t=l,δ).

The framework defined above may suffer from issues that pertain to recovering the correct solutions for the parameters being calibrated, also known as identifiability issues. These drawbacks are known in the literature and have been discussed in various works [50,51,52]. In this work, we limit our focus on improving the computational efficiency in a fully Bayesian formulation, while acknowledging this characteristic of the multifidelity framework.

### 2.2. Posterior Distribution

Assume a set of observations are available, namely, $D_{l}={\left\{x_{i},\theta_{i},y_{i}\right\}}_{i=1}^{N_{l}}$ and $D_{h}={\left\{x_{i},y_{i}\right\}}_{i=1}^{N_{h}}$ are the input to output sets of points corresponding to the low- and high-fidelity simulators, respectively. Conditioning the distribution of $y_{h}\left(x^{*},\theta \right)$ evaluated at some test point $x^{*}$ on the available data $D:=D_{l}\cup D_{h}$ and taking into account the prior choices and the independence between $y_{l}\left(\cdot \right)$ and δ(·), we can write the posterior density as a Gaussian process with mean and variance given by [46]
(4)$$\mu_{y_{h}}\left(x^{*},\theta \right)=t_{h}\left(x^{*},\theta \right)V_{h}^{-1}y$$
and
(5)$$\sigma_{y_{h}}^{2}\left(x^{*},\theta \right)=\sigma_{h}^{2}\left(x^{*}\right)-t_{h}\left(x^{*},\theta \right)V_{h}^{-1}t_{h}\left(x^{*},\theta \right).$$
In the above expressions, we use $y={\left(y_{l}^{T},y_{h}^{T}\right)}^{T}$,
(6)$$\begin{array}{c} V_{h}\left(\theta \right)=\left[\begin{array}{cc} V^{\left(l,l\right)} & V^{\left(l,h\right)}\left(\theta \right)\\ V^{\left(h,l\right)}\left(\theta \right) & V^{\left(h,h\right)}\left(\theta \right) \end{array}\right] \end{array}$$
where the diagonal block matrices are given by
(7)$$\begin{array}{c} \begin{array}{ccc} V^{\left(l,l\right)} & = & \sigma_{l}^{2}\left(R_{l}\left(D_{l}\right)+\sigma_{\epsilon_{l}}^{2}I\right),\\ V^{\left(h,h\right)}\left(\theta \right) & = & \sigma_{\delta }^{2}\left(R_{\delta }\left(D_{h}\right)+\sigma_{\epsilon_{h}}^{2}I\right)+\sigma_{l}^{2}\rho^{2}\left(R_{l}\left(D_{h}\left(\theta \right)\right)+\sigma_{\epsilon_{l}}^{2}I\right), \end{array} \end{array}$$
and $R_{t}\left(D_{t}\right)$ is the correlation matrix with entries $r_{t}\left(x,x^{\prime }\right)$ for x, $x^{\prime }\in D_{t}$, t=l,δ. In the above, $D_{h}\left(\theta \right):={\left\{\left(x_{i},\theta \right)\right\}}_{i=1}^{N_{h}}$ for $x_{i}\in D_{h}$. The off-diagonal blocks are written as
(8)$$V^{\left(l,h\right)}\left(\theta \right)=\rho V^{\left(l,l\right)}\left(D_{l},D_{h}\left(\theta \right)\right).$$
At last, we define the vector
(9)$$\begin{array}{c} t_{h}\left(x^{*},\theta \right)=\left(\begin{array}{c} \rho \sigma_{l}^{2}R_{l}\left(\left(x^{*},\theta \right),D_{l}\right)\\ \rho^{2}\sigma_{l}^{2}R_{l}\left(\left(x^{*},\theta \right),D_{l}\right)+\sigma_{\delta }^{2}R_{\delta }\left(x^{*},D_{h}\right) \end{array}\right). \end{array}$$

## 3. Variational Inference

Throughout this section we present the main ingredients of the variational inference framework for the purpose of training Gaussian process models by means of exploring a Bayesian posterior density. The target distribution in our case is the posterior distribution of the Gaussian process hyperparameters ω, defined as the set of length scales $\ell_{i,t}$, t=l,h along each dimension of X, the variance parameters $\sigma_{l}^{2}$, $\sigma_{h}^{2}$, $\sigma_{\epsilon_{t}}^{2}$, t,h, and the calibration parameters θ. This posterior density is conditioned on the training data D, which in general consist of the high- and low-fidelity input and output observations. From Bayes’ rule
(10)$$p\left(\omega |D\right)=\frac{p(D|\omega )p(\omega )}{p(D)}$$
the posterior density is known as a function of the likelihood term and the prior density, up to a proportionality constant. Variational inference [53,54] bypasses the challenge of sampling from the posterior, by approximating it by an element q(ω) chosen from a parametric family of distributions Q={q(ω|λ):λ∈Λ}, where Λ is some set that determines the parameterization of the densities in Q. The criterion for choosing the optimal density from the family is minimizing the Kullback–Leibler (KL) divergence between the candidate and the target densities. We define the KL divergence between the candidate and target densities as follows:(11)$$\mathrm{KL}\left[q\left(\omega |\lambda \right)||p\left(\omega \right)\right]=\int q\left(\omega |\lambda \right)\log\left(\frac{q(\omega |\lambda )}{p(\omega |D)}\right)d\omega .$$
Several techniques for solving the optimization problem exist in the literature [35] such as mean-field VI [55] or nonparametric VI [39], and they are typically tailored to problem-specific choices of prior densities, approximating family of distributions, and the inference problem under investigation.

One common characteristic of the approaches mentioned above is that they all transform the problem of minimizing the KL divergence to an equivalent maximization problem by substituting (Equation 10) into (Equation 11) to obtain
(12)logp(D)=KL[q(ω|λ)||p(ω)]+F[q],
where
(13)$$F\left[q\right]=H\left[q\right]+\int q\left(\omega |\lambda \right)\log\left(p(D,\omega )\right)d\omega$$
and H[q] is the entropy of q(ω|λ). Since the left-hand side of (Equation 12) is constant, we can conclude that the variational solution can be obtained by maximizing F[q], which is referred to as the *evidence lower bound (ELBO)*.

### Black-Box Variational Inference

One of the most popular choices for optimizing (Equation 13) is to directly employ a stochastic gradient descent or ascent algorithm, after observing that the objective function can be written as an expectation
(14)$$F\left[q\right]=E_{q}\left[\log p\left(D,\theta \right)-\log q\left(\omega |\lambda \right)\right],$$
where the expectation is taken with respect to q(ω|λ). The gradient of this expression with respect to the parameters λ that we seek to optimize is
(15)$$\nabla_{\lambda }F\left[q\right]=E_{q}\left[\nabla_{\lambda }\log q\left(\omega |\lambda \right)\left(\log p(D,\omega )-\log q(\omega |\lambda )\right)\right],$$
where the gradient $\nabla_{\lambda }\log q\left(\omega |\lambda \right)$ is known as the score function for any probability density *q* and the joint density can be expanded using Bayes’ rule to p(D,ω)=p(D|ω)p(ω). A Monte Carlo estimator of (Equation 15) can be written as
(16)$$\hat{\nabla_{\lambda }F}\left[q\right]=\frac{1}{N}\sum_{i=1}^{N}\nabla_{\lambda }\log q\left(\omega^{i}|\lambda \right)\left(\log p\left(D,\omega^{i}\right)-\log q\left(\omega^{i}|\lambda \right)\right),$$
where $\omega^{i}∼q\left(\omega |\lambda \right)$. Note that in the above expression, the gradient appears only on the score function, and can, in general, be computed analytically for certain families of distributions. On the contrary, the log-joint term logp(D,ω) which depends on the Bayesian model under investigation, needs not be differentiated. The gradient expression does not make any further assumptions and applies generically on every Bayesian inference problem, justifying the term coined to this approach as *black-box variational inference* [34].

To further scale the algorithm, we perform the log-joint function evaluations $p\left(D,\omega^{i}\right)=p\left(D|\omega^{i}\right)p\left(\omega^{i}\right)$ using batch sampling throughout the available dataset D, where each time, a random subset of the dataset is used to form the likelihood term. To put things in a realistic multifidelity context, it is highly unlikely that a big data problem will consist of a large number of high-fidelity observations. Therefore, in this work, we consider the following scenario where the number of training data points in $D_{l}$ is significantly larger that the number of high-fidelity observations $D_{h}$, that is $|D_{l}\left|\gg \right|D_{h}|$, thus, the batch sampling approach is applied only on $D_{l}$. At every evaluation of Equation (Equation 17), let $D_{l}^{i}$ be a random subset of $D_{l}$ and $D^{i}=D_{l}^{i}\cup D_{h}$, then Equation (Equation 17) is rewritten as follows:(17)$$\hat{\nabla_{\lambda }F}\left[q\right]=\frac{1}{N}\sum_{i=1}^{N}\nabla_{\lambda }\log q\left(\omega^{i}|\lambda \right)\left(\log p\left(D^{i},\omega^{i}\right)-\log q\left(\omega^{i}|\lambda \right)\right),$$
where $D_{l}$ is subsampled *N* times, that is, the number of Monte Carlo samples used to estimate $\hat{\nabla_{\lambda }F}\left[q\right]$. This scaling approaching was previously introduced in the literature as stochastic variational inference (SVI) [43].

## 4. Stochastic Optimization

### Manifold Gradient Ascent

For the case where the approximating family of distributions Q consists of multivariate Gaussian densities, that is, Q:={q(ω|λ):=N(ω|μ,Σ)}, a suitable optimization scheme needs to be employed over the parameters λ=(μ,Σ) such that the symmetric positive semidefiniteness property of the covariance matrix is not violated. Here, we employ a stochastic optimization scheme that is tailored particularly to our problem. The scheme applies a momentum algorithm for updating μ while performing the Σ update using a manifold gradient ascent step. For such a case, we make use of the natural gradient [56] as it is known to be invariant under parameterization [57].

The natural gradient on Riemannian manifolds is defined as
(18)$$\nabla_{\lambda }^{nat}F\left[q\right]=I_{F}^{-1}\nabla_{\lambda }F\left[q\right]$$
where $\nabla_{\lambda }F\left[q\right]$ is the regular gradient and $I_{F}$ is the Fisher information for density *q* that is defined as
(19)$$I_{F}\left(\lambda \right)=E_{q}\left[\nabla_{\lambda }\log q\left(\omega |\lambda \right){\left(\nabla_{\lambda }\log q\left(\omega |\lambda \right)\right)}^{T}\right].$$
In the Gaussian distribution case, the Fisher information matrix becomes
(20)$$\begin{array}{c} I_{F}\left(\mu ,\Sigma \right)=\left(\begin{array}{cc} \Sigma^{-1} & 0\\ 0 & I_{F}\left(\Sigma \right) \end{array}\right), \end{array}$$
where the elements of $I_{F}\left(\Sigma \right)$ are ${\left(I_{F}\left(\Sigma \right)\right)}_{\sigma_{ij},\sigma_{kl}}=\frac{1}{2}\mathrm{tr}\left(\Sigma^{-1}\frac{\partial \Sigma }{\partial \sigma_{ij}}\Sigma^{-1}\frac{\partial \Sigma }{\partial \sigma_{kl}}\right)$, and the inverse simplifies to
(21)$$\begin{array}{c} I_{F}{\left(\lambda \right)}^{-1}\approx \left(\begin{array}{cc} \Sigma & 0\\ 0 & \Sigma ⊗\Sigma \end{array}\right), \end{array}$$
where “⊗” is the Kronecker product. Finally, the natural gradient of F[q] can be written as
(22)$$\begin{array}{c} \begin{array}{ccc} \nabla_{\mu }^{nat}F\left[q\right] & = & \Sigma \nabla_{\mu }F\left[q\right]\\ \nabla_{\Sigma }^{nat}F\left[q\right] & = & \Sigma \nabla_{\Sigma }F\left[q\right]\Sigma \end{array}. \end{array}$$
In our stochastic gradient ascent scheme, the parameters μ are updated using a momentum algorithm with updating step
(23)$$\mu_{t+1}=\mu_{t}+\gamma m_{\mu_{t}}$$
where the momentum term $m_{\mu_{t}}$ is given by
(24)$$\begin{array}{c} m_{\mu_{t+1}}=\upsilon m_{\mu_{t}}+\left(1-\upsilon \right)\nabla_{\mu }^{nat}F\left[q\right]. \end{array}$$
For the update on Σ, it is necessary to map the point on the tangent space, indicated by the steepest ascent direction, back to the manifold. For that, we use a *retraction mapping* that approximates the exponential map of the manifold of symmetric positive semidefinite matrices [58].

In our case, we use
(25)$$R_{\Sigma }\left(\xi \right)=\Sigma +\xi +\frac{1}{2}\xi \Sigma^{-1}\xi .$$
Further, for the momentum update on the manifolds, we apply a *vector transport* that further projects the translated points back to the tangent space, as was first done in [59]. For our purposes, we apply the following mapping:(26)$$\Gamma_{\Sigma_{1}\to \Sigma_{2}}\left(\xi \right)=U\xi U^{T},\;\;U={\left(\Sigma_{2}\Sigma_{1}^{-1}\right)}^{1/2}.$$
Finally, our computational algorithm is summarized in Algorithm 1.
**Algorithm 1:** Manifold gradient ascent
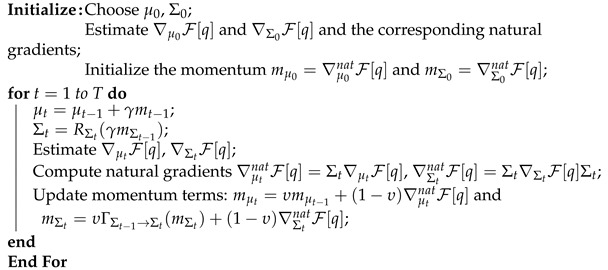


## 5. Numerical Examples

We studied the performance of the proposed algorithm on three problems. One metamodeling problem and two multifidelity model calibration problems are used in the sections that follow.

### 5.1. Academic Example

We first considered the following mathematical functions
(27)$$\begin{array}{c} \begin{array}{ccc} f_{1}\left(x,\theta \right) & = & \theta_{1}\left(8w^{T}x-2\right)\sin\left(5w^{T}x-4\right)+\theta_{2}\left(2w^{T}x+\frac{1}{2}\right)\\ f_{2}\left(x,\theta \right) & = & f_{1}\left(x,\theta \right)+30{\left(w^{T}x\right)}^{2}, \end{array} \end{array}$$
with the coupling indicating that $f_{1}\left(x,\theta \right)$ can be considered to be a low-fidelity simulator and $f_{2}\left(x,\theta \right)$ the high-fidelity function. We took $x\in R^{10}$ and the vector w was considered a set of known parameters projecting the 10-dimensional vector x to R. For this example, we took
(28)$$\begin{array}{c} w=\left[\begin{array}{c} 0.14042\\ -0.35474\\ 0.42674\\ -0.09312\\ -0.21463\\ 0.26425\\ 0.25603\\ -0.18959\\ 0.00467\\ -0.66800 \end{array}\right]. \end{array}$$
A set of $10^{4}$ training points was generated from the low-fidelity function, that is, $D_{l}={\left\{x_{i},\theta_{i},y_{i}\right\}}_{i=1}^{10^{4}}$ while $D_{h}={\left\{x_{i},y_{i}\right\}}_{i=1}^{200}$ consisted of 200 points simulated from $f_{2}$, where the calibration parameters were fixed to θ=(3/2,30). All inputs were generated using a uniform Latin hypercube sampling on ${\left[-2,2\right]}^{10}$, while the $\theta_{i}$’s were sampled uniformly within [0.5,2.5]×[20,40]. The data are shown in Figure 1.

To test for the robustness of the approach, we first performed the ELBO optimization corresponding to training an autoregressive GP model on the available training data using a varying number of Monte Carlo samples used to evaluate the ELBO gradient estimate (Equation 17), namely N=10,50, and 100. We ran $5\times 10^{3}$ iterations of algorithm 1 using an initial learning rate $\gamma_{0}=0.0001$, momentum weight parameters υ=0.6, and a random batch size equal to 50 data points (0.5% of the full dataset) to enable the SVI feature. As expected, the runtimes scaled linearly from 13 min for N=10 to 61 min for N=50, and 125 min for N=100. Improving the quality of the MC estimate of the F[q] function was expected to improve optimization performance. This is highlighted in Table 1 which shows the final ELBO values F[q] achieved after the optimization procedure was over and the mean predictive standard deviation (std) over all test points. In addition, Figure 2 shows the convergence of the objective function. It can be observed that increasing *N* resulted in more robust convergence and attaining higher values closer to the true optimum that subsequently resulted in better posterior estimates. The decrease in mean predictive std indicated that the predictive accuracy was improved as well and the confidence increased.

Next, Figure 3 shows the comparison between the observations and the trained model predictions along with a 45-degree line plot for the case where the number of MC samples was as low as 10. As can be seen, the red “•” marks that correspond to the discrepancy-adjusted prediction match exactly the observations and the variance remains low. The blue “×” marks corresponding to the inferred low-fidelity simulator η(x,θ) fall below the line, which agrees with the observed trends of the true functions as seen in Figure 1. Specifically, the low-fidelity function appears to be the closest possible to the high-fidelity one on design points x corresponding to values of $w^{T}x$ near the origin, which is when we should expect the discrepancy term points to be the closest to the 45-degree line. When η(x,θ) reaches very low or very high values (near −200 or 200, respectively), the discrepancy is the largest, and indeed the points are far from the 45-degree line.

Figure 4 shows the prediction versus observations plots for 500 test data points along with a 45-degree line plots again for the three cases where the number of MC samples was 10, 50, and 100, respectively. At last, Figure 5 shows the posterior densities of the two calibration parameters $\theta =(\theta_{1},\theta_{2})$ obtained using the VI framework. We observe a clear improvement in the accuracy of the $\theta_{1}$ estimate as the number of Monte Carlo samples increase from 10. To ensure numerical stability in our implementation, the Gaussian approximation was applied on the logθ and the resulting density plots were based on a kernel density estimation using $5\times 10^{3}$ samples from the optimal log-normal approximation that was obtained using the VI approach.

### 5.2. Chicago Crimes Statistics Dataset

In this section, we demonstrate the applicability of the proposed approach on a metamodeling task. The dataset used for this problem was one of the three datasets under the *Query Analytics Workloads Dataset* section, hosted by the University of California Irvine open-source machine learning data repository (https://archive.ics.uci.edu/ml/datasets/Query+Analytics+Workloads+Dataset accessed on 31 May 2022). This dataset was used in other recent works [60,61] in order to benchmark the performance of the proposed novel machine learning algorithms and has been derived from synthetic query analytics workloads from (https://data.cityofchicago.org/Public-Safety/Crimes-2001-to-Present/ijzp-q8t2 accessed on 31 May 2022). The quantity of interest being modeled was the number of crimes reported or simply the count of crimes in a particular region, in the city of Chicago [60]. The variables used to define the region included the *x* and *y* coordinates of the center of the region and the radius of the region. Thus, the problem had three inputs and one output. The dataset had ten thousand pairs of inputs and outputs. We leveraged nine thousand points for training the fully Bayesian metamodel and left out one thousand points as *test data* in order to evaluate the predictive performance of the trained model and we ran $2\times 10^{4}$ iterations of our optimization scheme.

Two clear observations from Figure 6 are: (a) the predictive performance visibly improves as the batch size of subsampled training data increases from across the three subfigures (here we used batch sizes equal to 45, 67, and 90 samples that corresponded to using 0.5%, 0.75%, and 1% of the available dataset), and (b) the predictive epistemic uncertainty of the trained model also decreases, indicating a higher confidence in the model. Table 2 shows the improvement of the predictive accuracy of the model as the batch size used for training is increased, as that is illustrated through the root-mean-squared error (RMSE) and the mean std values.

In addition to these, Figure 7 shows the increase in runtime of the algorithm as the batch size of subsampled training data increases. For reference, we also present the runtimes of the sparse GP implementations presented in [4] using the GPy package [62] for the same number of iterations and batch size and a latent variable with 80 data points. As can be seen, our approach reduced the runtime significantly for very small batch sizes while the performance of the two algorithms was about the same when batch size became 90.

### 5.3. Torsional Vibration Problem

We considered the torsional vibration problem on the system depicted in Figure 8 consisting of three shafts and two discs of varying geometric characteristics and elasticity properties. Our goal was to built a Gaussian process metamodel on the quantity of interest that expressed the lowest natural frequency, given as
(29)$$Y=\sqrt{\frac{-b-\sqrt{b^{2}-4ac}}{2}}/2\pi ,$$
where a=1,
(30)$$\begin{array}{cc} b=-\left(\frac{K_{1}+K_{2}}{J_{1}}+\frac{K_{2}+K_{3}}{J_{2}}\right), & c=\frac{K_{1}K_{2}+K_{2}K_{3}+K_{1}K_{3}}{J_{1}J_{2}}. \end{array}$$

The torsional stiffness values were given by
(31)$$K_{i}=\theta_{1}\frac{\pi G_{i}d_{i}}{32L_{i}},\;\;i=1,2,3$$
and the polar moments of inertia were given by
(32)$$J_{j}=\theta_{2}M_{j}{\left(\frac{D_{j}}{2}\right)}^{2},\;\;i=1,2$$
with $M_{j}=\frac{\rho_{j}}{g}\pi t_{j}\frac{D_{j}}{4}$, j=1,2. We considered a high-fidelity simulator where $Y_{hf}$ was evaluated using $\theta_{1}=\pi /32$, $\theta_{2}=\frac{1}{2}$ and shaft diameters $d_{1}=2$, $d_{2}=1.825$, and $d_{3}=2.25$ in expressions (Equation 31) and (Equation 32), while data from a low-fidelity $Y_{lf}$ were also used, where $\theta_{1}$ and $\theta_{2}$ were considered unknown parameters to be inferred and all diameters were taken equal $d_{1}=d_{2}=d_{3}=2$. All 12 remaining geometric and elasticity properties of the system were assumed to be design parameters and are described in Table 3.

We considered again an experimental scenario in the big data regime, where $10^{4}$ simulation data points were generated from $J_{1}$ and a much smaller number of high-fidelity observations were available from $J_{2}$. We tested the robustness of the approach by varying the number of high-fidelity observations from only 50 points up to 250 and we compared the runtimes. Due to the increasing number of data points used to optimize the ELBO, it became necessary to adjust the maximum number of iterations for which the optimization algorithm ran, and therefore, the resulting runtime was affected. For the first three cases, we performed 1000 iterations; for the case $N_{hf}=200$, we performed 1500 iterations; and for the remaining case ($N_{hf}=250$), 2000 iterations were found to be necessary. Figure 9 shows the convergence of the ELBO function along with the root-mean-squared error (RMSE) values obtained for each trained model, based on 100 test data points. As expected, the RMSE goes down with an increasing number of high-fidelity data as shown in Figure 9 bottom.

The posterior results for the calibrated parameters along with the runtime for each case are shown in Table 4. As can be seen, the true values (0.98 and 0.5) fall within the reported mean values of θ ± 2 standard deviations for all cases. At last, the comparison of the model prediction versus observation, along with the 45-degree line plots, is provided for the worse and best cases ($N_{hf}=50,250$) in Figure 10.

## 6. Conclusions

We enhanced and extended the state-of-the-art stochastic variational Bayesian formulation for tasks that use GPs for multifidelity metamodeling and model calibration tasks, in order to treat problems with tens of thousands of training data and model calibration problems with more than ten inputs. The proposed mathematical formulation extended two classic approaches, the so-called black-box VI and stochastic VI, while utilizing a manifold gradient ascent scheme to accomplish the task of inferring the GP hyperparameters as well as the calibration parameters. The major impact of our work was being able to perform a fully Bayesian uncertainty quantification while training and calibrating models using multifidelity GPs, albeit with large datasets and a moderately large number of inputs. Numerical results on two challenging engineering problems visibly demonstrated a scale up of classical Bayesian GPs for multifidelity modeling to calibrate *untuned* computer simulators, by enabling savings in computation. This *speed-up* is critical for engineering applications, especially in the industry, where repeated model calibration tasks are a common occurrence and can lead to accumulated savings using the proposed approach.

This work showed promise for accelerating the training procedure in Gaussian process based metamodels without relying on enormous computational power. The key characteristic in our approach was the batch-sampling step used in the stochastic variational inference framework, which allowed the fast computation of the likelihood term and accelerated the optimization task. One key challenge in our approach is that fine tuning of the optimization is required in order to ensure a sufficiently large updating step in the optimization scheme, while at the same time avoiding overshooting. Fine tuning the algorithm heavily depends on the size of the batch samples being used, which is also relative to the original data size that is available. Extremely small batch samples can result in very inaccurate likelihood evaluations and eventually miss the optimum. Another important aspect mentioned above is the number of Monte Carlo samples used for approximating the ELBO function. Very small number of samples can lead to inaccurate estimates with large variance that fail to converge, while on the other hand, a high number of samples makes the algorithm computationally expensive and fails to achieve the desired speed up. Typically, big data problems in Bayesian inference exhibit a well-defined posterior, therefore optimizing the ELBO should always be a feasible task given that some fine tuning has been performed. A limitation of the approach would the case where a big part of the data set is corrupted or contains high noise, in which case, the exploration of the posterior via VI might become challenging due to the complex nature of the true posterior. In such cases, more complex variational approximations need to be considered which could, however, make the algorithm less computationally efficient.

Other general challenges, not associated specifically with our approach, are problems of extremely high input and output dimensions as well as highly nonsmooth response functions. In such cases, a further development of our framework might be necessary such that it aligns with similar approaches in the literature, for instance, enabling covariance matrix sparsity, employing nonsmooth correlation kernels and last, but not least, leveraging parallel computing.

Directions for future work include scaling up the proposed approach to problems with higher input dimensionality, i.e., hundreds of inputs and with more than one sources of information with lower-fidelity and large training data. Additionally, the proposed approach needs more work in order to be applied to problems where the different sources do not share the same inputs.

## Figures and Tables

**Figure 1 entropy-24-01291-f001:**
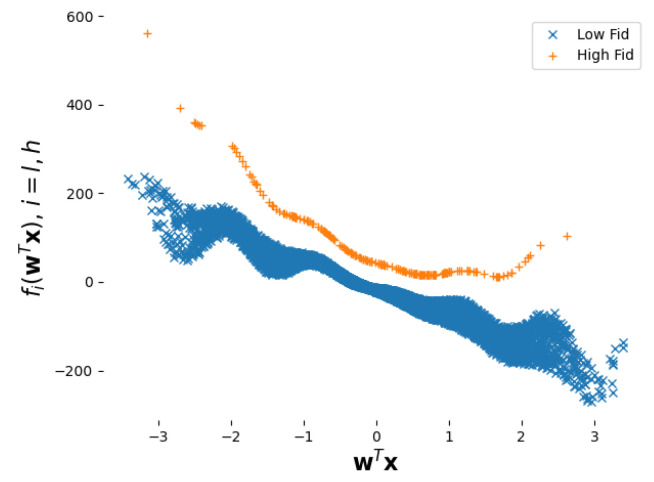
Training data for the academic example. Low-fidelity data are depicted with blue “×” while high-fidelity observations are depicted with orange “+”.

**Figure 2 entropy-24-01291-f002:**
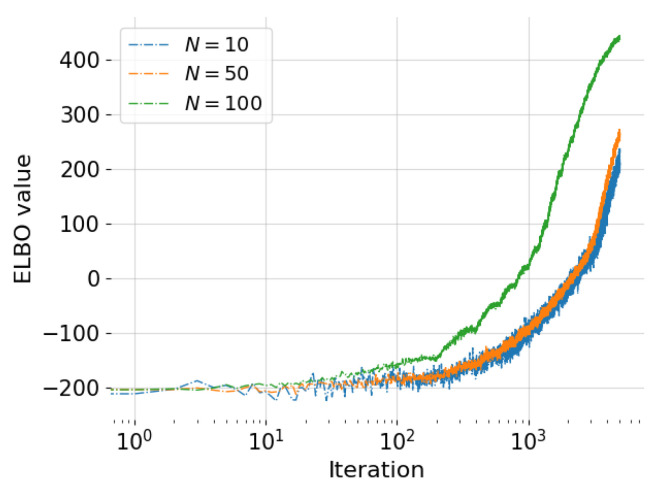
Academic example: ELBO values vs. iterations for each training case (N=10,50, and 100).

**Figure 3 entropy-24-01291-f003:**
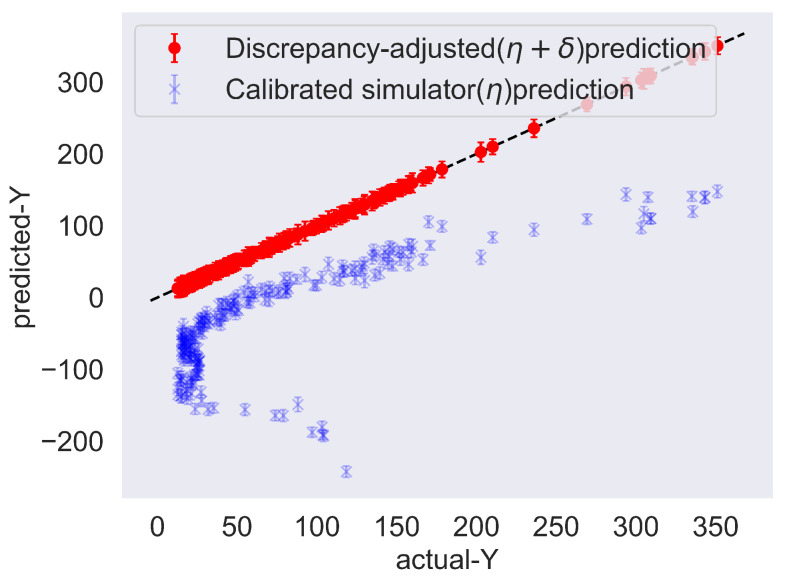
Prediction on the training data for low-fidelity term η(x,θ) and discrepancy-adjusted high-fidelity output $y_{h}\left(x,\theta \right)$ versus observations. Model was trained using 10 MC samples for the ELBO evaluation.

**Figure 4 entropy-24-01291-f004:**
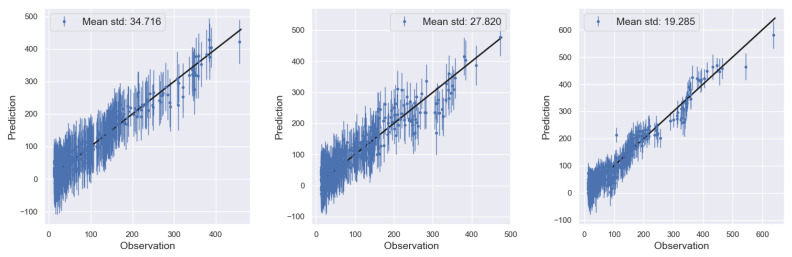
Prediction versus observations for 500 test data points. The model was trained using 10 MC samples for the ELBO gradient evaluation.

**Figure 5 entropy-24-01291-f005:**
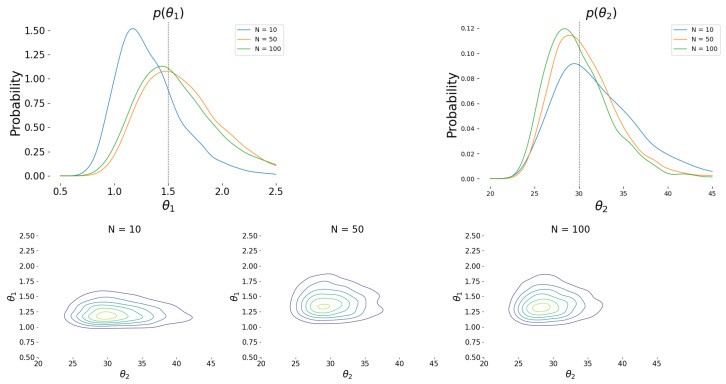
**Top**: Posterior marginal densities for $\theta_{1}$ (**left**) and $\theta_{2}$ (**right**) obtained after the ELBO optimization with a varying number of Monte Carlo samples. **Bottom**: Joint density plots obtain for N=10,50,100.

**Figure 6 entropy-24-01291-f006:**
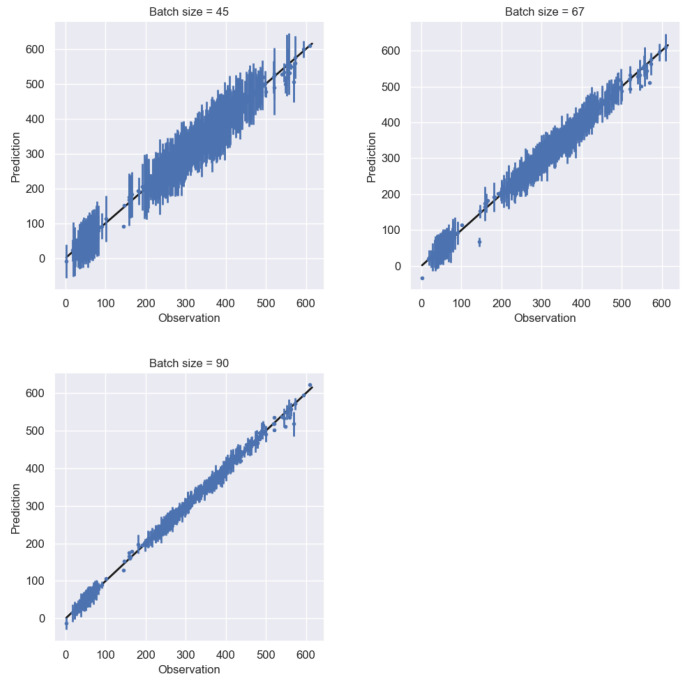
Prediction versus observations for 1000 test data points. The three models were trained using batch sizes equal to 45 (**top left**), 67 (**top right**), and 90 (**bottom**) samples that were resampled from the full dataset.

**Figure 7 entropy-24-01291-f007:**
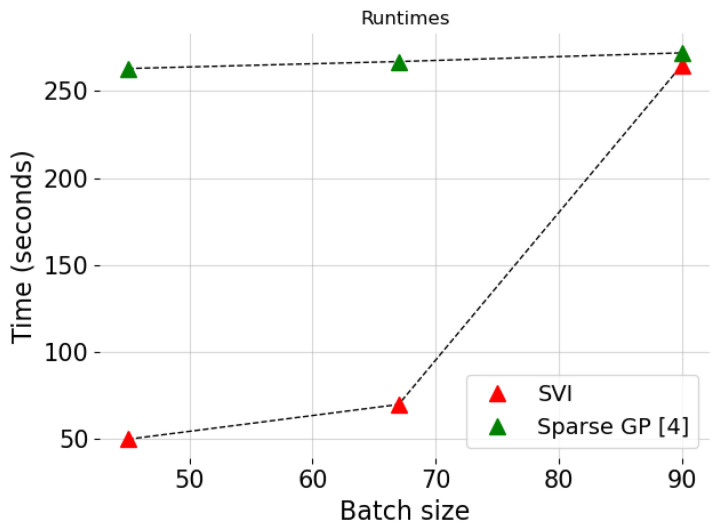
Runtime comparison for three different batch sizes for the Chicago crimes dataset.

**Figure 8 entropy-24-01291-f008:**
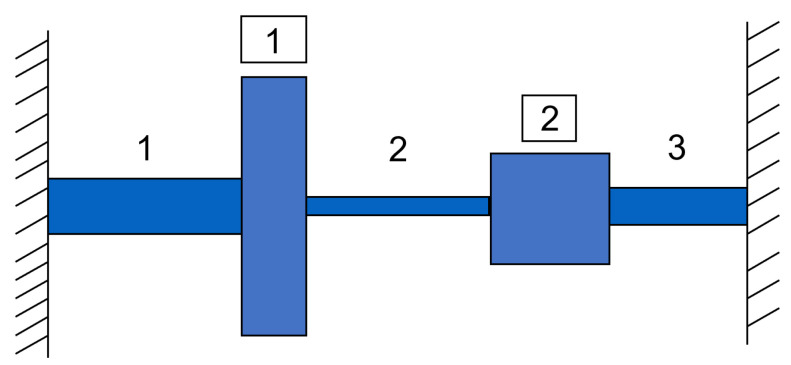
Torsional vibration on system consisting of and three shafts and two discs placed in between successive shafts.

**Figure 9 entropy-24-01291-f009:**
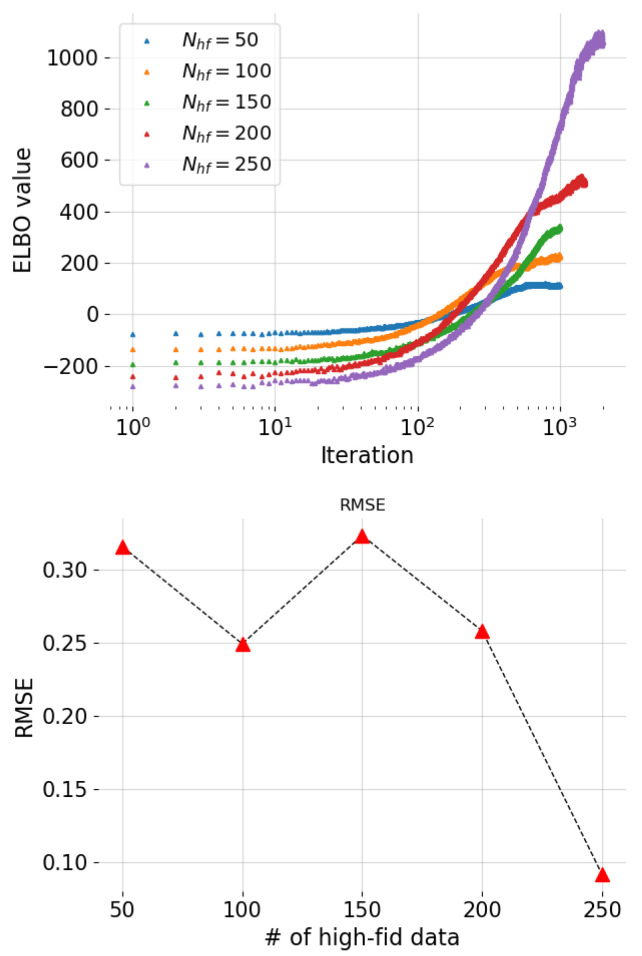
Torsional vibration problem: plots of the ELBO function vs. number of iterations (**top**) and plot of the RMSE values (**bottom**) for different numbers of high-fidelity data points $N_{hf}$.

**Figure 10 entropy-24-01291-f010:**
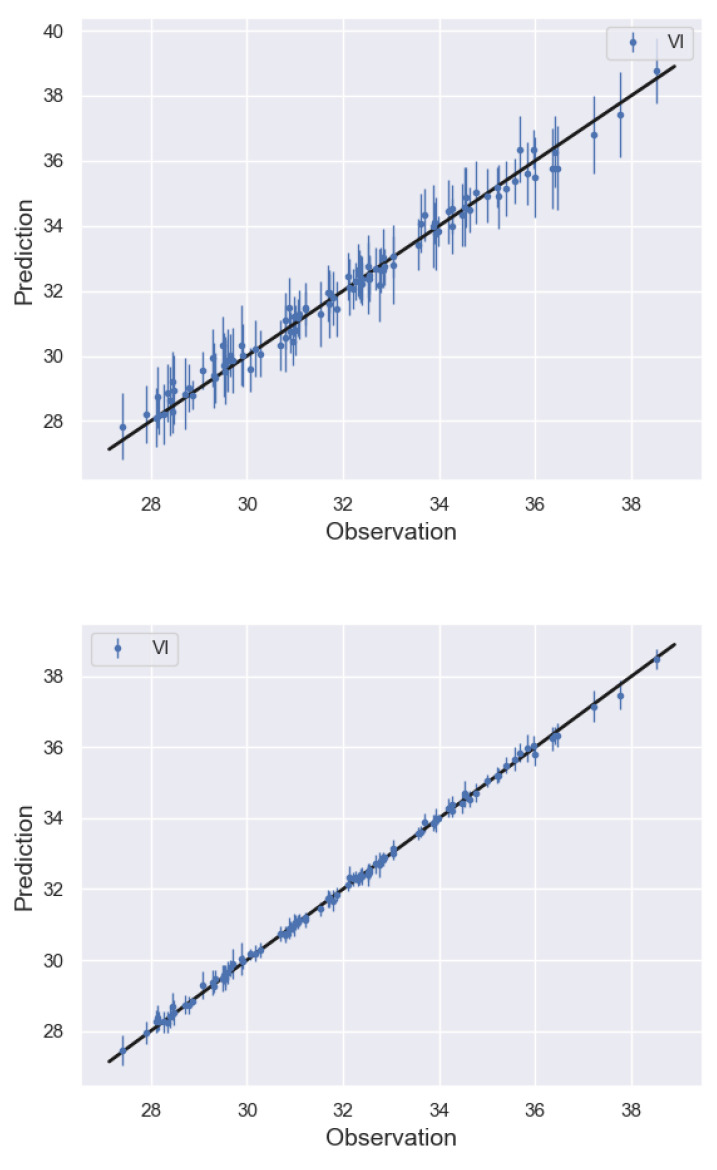
Torsional vibration problem: comparison of trained model prediction vs. observation on 100 test data points along with 45-degree line plots. The high-fidelity points used to train the model were $N_{hf}=50$ (**top**) and $N_{hf}=250$ (**bottom**).

**Table 1 entropy-24-01291-t001:** Academic example: final ELBO values after 5000 iterations and the computational runtimes for each training case (N=10,50, and 100).

*N*	Final F[q]	Mean Predictive Std	Runtime
10	228.77	34.71	13’
50	265.64	27.82	61’
100	439.75	19.28	125’

**Table 2 entropy-24-01291-t002:** Chicago crimes dataset: root-mean-squared error (RMSE) values and mean standard deviations for the resulting model trained using different batch sizes.

Batch Size	RMSE Value	Mean Predictive Std
45	6.445	24.626
67	6.315	15.065
90	5.254	7.889

**Table 3 entropy-24-01291-t003:** Torsional vibration problem: description of the 12-dimensional input parameters and their values ranges. Length, diameters and thicknesses are given in inches, moduli of rigidity are in lb/sq inch, and weight densities are expressed in lb/cubic inch.

Part	Parameter	Value Range
Shaft 1	Length $L_{1}$	[9, 11]
	Modulus of rigidity $G_{1}$	[1053, 1287] $\times \;10^{5}$
Shaft 2	Length $L_{2}$	[10.8, 13.2]
	Modulus of rigidity $G_{2}$	[558, 682] $\times \;10^{4}$
Shaft 3	Length $L_{3}$	[7.2, 8.8]
	Modulus of rigidity $G_{3}$	[351, 429] $\times \;10^{4}$
Disk 1	Diameter $D_{1}$	[10.8, 13.2]
	Thickness $t_{1}$	[2.7, 3.3]
	Weight density $\rho_{1}$	[0.252, 0.308]
Disk 2	Diameter $D_{2}$	[12.6, 15.4]
	Thickness $t_{2}$	[3.6, 4.4]
	Weight density $\rho_{2}$	[0.09, 0.11]

**Table 4 entropy-24-01291-t004:** Torsional vibration problem: posterior statistics for the calibration parameters $\left(\theta_{1},\theta_{2}\right)$ and the computational runtimes for each training case.

$N_{\mathrm{hf}}$	$\theta_{1}$ (Mean, Std)	$\theta_{2}$ (Mean, Std)	Runtime
50	(0.092, 0.01)	(0.484, 0.042)	9.7’
100	(0.091, 0.01)	(0.487, 0.111)	12.9’
150	(0.132, 0.03)	(0.682, 0.179)	14.1’
200	(0.145, 0.27)	(0.554, 1.614)	44.6’
250	(0.088, 0.0008)	(0.450, 0.0007)	54.4’

## Data Availability

All details necessary for regenerating the data used in numerical examples Section 5.1 and Section 5.3 is provided in each problem description. Links for the publicly available dataset used in numerical example Section 5.2 is also provided in the relevant section.

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
