# Peer review of "Multifidelity Model Calibration in Structural Dynamics Using Stochastic Variational Inference on Manifolds"

_entropy, 2022, doi:10.3390/e24091291_

Round 1
Reviewer 1 Report
For the sake of completeness and to be accessible to a wider audience, it is very important that the term 'fidelity' must be defined and briefly explained. This can be done in the second paragraph. Also, the paper seems to use the terms single and multi and low and high, interchangeably. This is confusing. Please be consistent and clear in your use of the terms.
Author Response
We are thankful to the reviewer for the comment above. We have extended the second paragraph in the introduction (as suggested) with details on how the multi-fidelity simulation context is defined, followed by relevant references. We hope that this fully address the comment and clarifies any confusion.
Reviewer 2 Report
The manuscript under consideration introduces a fully Bayesian framework for Gaussian process modeling employing multiple fidelities of training data and stochastic variational inference to enable its efficacy in large data scenarios. The presented framework is useful and relevant to industrial application, but several presentation issues must be addressed prior to publication.
1) There are some grammatical issues in the manuscript that need attention
2) On page 8 it is not clear how the number of MC samples affects prediction accuracy and uncertainty. A figure or table would be helpful to clarify.
3) All parity plots (e.g. Figure 3) should have a square aspect ratio to aid in readability
4) The subplots in Fig. 4 should be rearranged to aid in readability
5) The uncertainty in Fig. 5 appears to increase between batch sizes of 45 and 67. It would be helpful to have a table showing mean prediction uncertainty and error versus batch size.
6) How were the batch sizes in Sec 5.2 selected? The values seem rather arbitrary. This indicates that considerable tuning was required before narrowing down on these particular batch sizes. If this was the case, it should be discussed in the manuscript.
7) The log-x scaling in Fig. 8 (left) makes it challenging to judge convergence.
8) Please remark on whether the implementation will be made available to the community. This would dramatically increase the impact of this work if so. If not, please provide justification.
Author Response
We are grateful for the feedback we have received from this reviewer as it allowed us to improve the readability of our manuscript and to clarify certain points that can better illustrate the applicability of our proposed methodology.
Below the reviewer can find the details on how we have addressed his/her concerns and our answers to the questions and we hope that this revision meets his/her expectations for publication.
1) There are some grammatical issues in the manuscript that need attention.
We have taken a thorough pass on the manuscript and have corrected several grammatical errors where those appeared.
2) On page 8 it is not clear how the number of MC samples affects prediction accuracy and uncertainty. A figure or table would be helpful to clarify.
The number of MC samples improves the quality of the ELBO estimate and therefore the optimization performance that will result in better posterior estimates which consequently will improve prediction. We have added a table
showing the resulting objective function values as well as the mean predictive std value and a figure with the ELBO convergence. In addition, Fig 4 now contains all three 45-degree line plots where it can be seen that the predictive
accuracy improves significantly for N = 100.
3) All parity plots (e.g. Figure 3) should have a square aspect ratio to aid in readability
We have replaced the 45-degree line plots for numerical example 1 & 2 with ones that have squared aspect ratio. We find that the ones corresponding to the last vibration torsion example (Fig. 10) are in large enough font with clear
readability and need no replacement.
4) The subplots in Fig. 4 should be rearranged to aid in readability
The feedback has been incorporated.
5) The uncertainty in Fig. 5 appears to increase between batch sizes of 45 and 67. It would be helpful to have a table showing mean prediction uncertainty and error versus batch size.
This appears to be an artifact of the poor estimator used for the objective function evaluation. We repeated the simulations with an increased size of MC samples used to evaluate the ELBO function (20 as opposed to 10 used
before) and we noticed that the convergence is more consistent and the errorbars gradually decrease this time. We have included a table with predictive accuracy (root mean squared error) and mean std values for comparison.
6) How were the batch sizes in Sec 5.2 selected? The values seem rather arbitrary. This indicates that considerable tuning was required before narrowing down on these particular batch sizes. If this was the case, it should be discussed
in the manuscript.
The batches sizes correspond to certain percentage ratios of the full training dataset. Specifically we used a 0.5%, 0.75% and 1% of the 9000 available training points as the batch size which gives 45, 67 and 90 respectively. We have
added this clarification in the manuscript as well.
7) The log-x scaling in Fig. 8 (left) makes it challenging to judge convergence.
Our focus here is primarily on illustrating how the performance of the trained model improves as a function of the number of high fidelity points used in the training procedure. The ELBO functions in this Figure seem to have started
converging near iteration 1000 (the curves have started becoming flat) but indeed it is hard to assess convergence. We
find that for the sake of comparison, 1000 iterations are good enough and we need not use strict convergence criteria
as a stopping rule in our optimization scheme.
8) Please remark on whether the implementation will be made available to the community. This would dramatically increase the impact of this work if so. If not, please provide justification.
We will try to release a version at a latter date, as at this time the implementation involves some in-house GE software that would need additional work to be sanitized for public release.
Reviewer 3 Report
This is a very interesting and well written article. I especially liked the use of the realistic examples.
Author Response
We deeply appreciate the reviewer’s positive feedback on our article and the acknowledgement of the applicability of the approach on real-world industrial problems.
Round 2
Reviewer 1 Report
The authors have made the changes I had requested.